# Improving Intervention Efficacy via Concept Realignment in Concept Bottleneck Models

**Nishad Singhi**[1]    **Jae Myung Kim**[1]    **Karsten Roth**[1]    **Zeynep Akata**[2]
[1]Tübingen AI Center, University of Tübingen    [2]Helmholtz München, TU München
nishadsinghi@gmail.com

## Abstract

Concept Bottleneck Models (CBMs) ground image classification on human-understandable concepts to allow for interpretable model decisions. Crucially, the CBM design inherently allows for human interventions, in which expert users are given the ability to modify potentially misaligned concept choices to influence the decision behavior of the model in an interpretable fashion. However, existing approaches often require numerous human interventions per image to achieve strong performances, posing practical challenges in scenarios where obtaining human feedback is expensive. In this paper, we find that this is noticeably driven by an independent treatment of concepts during intervention, wherein a change of one concept does not influence the use of other ones in the model's final decision. To address this issue, we introduce a trainable concept intervention realignment module, which leverages concept relations to realign concept assignments post-intervention. Across standard, real-world benchmarks, we find that concept realignment can significantly improve intervention efficacy; significantly reducing the number of interventions needed to reach a target classification performance or concept prediction accuracy. In addition, it easily integrates into existing concept-based architectures without requiring changes to the models themselves. This reduced cost of human-model collaboration is crucial to enhance the feasibility of CBMs in resource-constrained environments.

## 1 Introduction

Despite tremendous progress in the performance of Deep Learning (DL) techniques on challenging benchmarks such as ImageNet (Russakovsky et al., 2015), their adoption to high-stakes scenarios such as healthcare has been limited (Koh et al., 2020; Zarlenga et al., 2022) This is in large part due to unpredictable biases and failure cases of deep models when transferring to unseen data or complex & ambiguous cases grounded in the numerous model parameters, architecture designs and training choices (Dwork et al., 2012; Chouldechova, 2016; Geirhos et al., 2020; Eulig et al., 2021; Mehrabi et al., 2022; Dullerud et al., 2022; Brown et al., 2023; Roth et al., 2024; Casper et al., 2024). The black-box nature of typical DL models and their representation spaces (Shwartz-Ziv & Tishby, 2017; Locatello et al., 2019; Buhrmester et al., 2019; Roth et al., 2023; Casper et al., 2024) further exacerbate this problem, as it makes understanding and debugging the decision-making process of these models difficult. To foster trust and adaptation to new contexts, Koh et al. (2020) introduced Concept-Bottleneck Models (CBMs) which ground model decision process on human-interpretable concepts, while also allowing users to provide additional context at test time to update the final model prediction. In particular, CBMs operate by first predicting a set of human-interpretable concepts from an image, before predicting the label from these concepts. While simple, this architecture aligns the features used in the model decision process with those used by humans for the same task, allowing users to understand the model's behavior in terms of concepts they are familiar with. Moreover, the use of human-interpretable concepts implies that users can verify and even update the predictions of individual concepts, leading the model to update its final prediction to improve classification accuracy. While this allows for human involvement, it's essential to note that human annotation is expensive. Therefore, ideally, models should be able to improve with minimal human input. However, in many cases, numerous interventions may be needed for each image to significantly boost the model's accuracy. For example, in the widely used CUB benchmark (bird

classifications, Wah et al. (2011)), it takes about 13 interventions per image on average to raise the accuracy of a concept-based model from 68% to 90%. A large part of this limited intervention efficacy can be traced down to the independent nature of concept interventions, wherein each concept is updated separately. This means that correcting for one concept does not affect which other concepts are predicted for the same image. However, concepts in real life are often correlated, and informing the model about one concept should consequently influence the use of related ones. Not doing so implies suboptimal use of human feedback during inference.

In this paper, we study the extent of this crucial aspect when operating with concept-based models. Our study highlights how the use of a **simple *concept intervention realignment* module**, which learns from statistical concept relations, can effectively and automatically realign concept values after some interventions have been performed. Our experiments reveal how our concept intervention realignment can seamlessly integrate into and improve any existing concept-based approach (e.g. default CBMs Koh et al. (2020), advanced CEMs Zarlenga et al. (2022)). Across three standard, real-world benchmarks (CUB Wah et al. (2011), CelebA Liu et al. (2015) and AwA2 Xian et al. (2018)), we showcase consistent, in parts very significant improvements in intervention efficacy.

## 2 BACKGROUND

### 2.1 CONCEPT-BOTTLENECK MODELS

A Concept-Bottleneck Model (CBM), $h = f(g(x)) : \mathcal{X} \to \mathcal{Y}$, can be viewed as a composition of two models: the concept extractor $g : \mathcal{X} \to \mathcal{C}$, and the classification head $f : \mathcal{C} \to \mathcal{Y}$. $\mathcal{X}, \mathcal{Y}$, and $\mathcal{C}$ are the sets of inputs, class labels, and concepts, respectively. First, the concept extractor predicts the concepts in the input $x \in \mathbb{R}^d$ as $\hat{c} = g(x) \in \mathbb{R}^k$, after which the classifier predicts the label of the image using the predicted concepts as $\hat{y} = f(\hat{c}) \in \mathbb{R}^M$ with $d, k$, and $M$ the number of input dimensions, concepts, and classes, respectively. Since their introduction by Koh et al. (2020), follow-up works have tried to improve CBMs and overcome their limitations. Concept Embedding Models (Zarlenga et al., 2022) were proposed to generalize this framework, with concepts represented by an embedding vector as opposed to a scalar probability value, increasing task performance while maintaining interpretability. To mitigate the need for explicit concept supervision during training, recent works have attempted to use pre-trained vision backbones and language guidance to build interpretable models without requiring concept annotations in the training data (Oikarinen et al., 2022; Yang et al., 2023; Yuksekgonul et al., 2022). Kim et al. (2023) proposed probabilistic CBMs to model uncertainty in concepts and the final class predicted by the model. While these works have focused on improving the architecture of CBMs, our work is orthogonal to them and enables these models to correct their predictions using fewer human interventions at test time.

### 2.2 INTERVENTIONS ON CBMS

At test time, the user can intervene on the set of concepts $\mathcal{S} \subseteq \{1, 2, ..., k\}$, and change the set of concept predictions from $\hat{c}$ to $\tilde{c} = \{c_{\mathcal{S}}, \hat{c}_{\backslash \mathcal{S}}\}$, consequently updating the final prediction from $\hat{y}$ to $\tilde{y} = f(\tilde{c})$, where $c_{\mathcal{S}}$ are the ground truth values of the intervened concepts and $\hat{c}_{\backslash \mathcal{S}}$ are the model's predictions of non-intervened concepts. In the original CBM paper, Koh et al. (2020) demonstrated that selecting concepts randomly, and correcting them if needed, improved classification performance. Subsequently, Chauhan et al. (2023) and Sheth et al. (2022) proposed uncertainty-based strategies to select which concepts an expert should intervene on. Shin et al. (2023) conducted an in-depth study on various strategies to select concepts with an emphasis on task performance and the cost of executing the strategies. Zarlenga et al. (2023) introduced interventions at training time to improve the receptiveness of models to test-time interventions. All of these works assume concepts to be independent, which drives the aforementioned lack of intervention efficacy. Our work is *complementary to these approaches*, as it aims to update the predictions of all concepts when an expert intervenes on *any subset of concepts*.

## 3 CONCEPT INTERVENTION REALIGNMENT MODULE

Existing methods do not account for the relationships between concepts during test-time interventions, leading to suboptimal usage of human feedback. To deal with this, we propose a *concept*

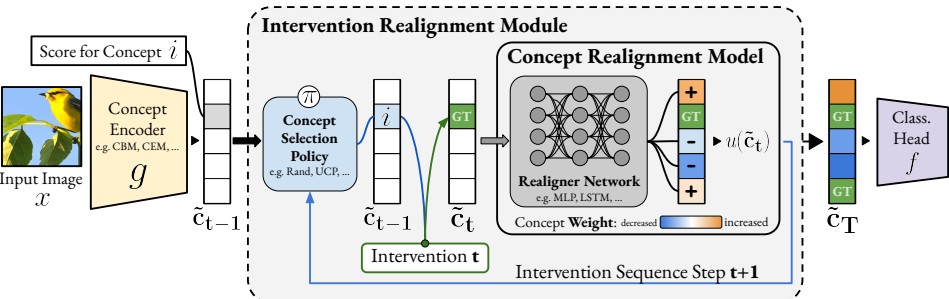

Figure 1: **Illustration of the concept intervention realignment module.** Given the concept encoding $g(x)$, we intervene on the concept $i$ selected by a concept selection policy $\pi$. This concept is replaced with a ground-truth (GT) value ($\in \{0, 1\}$ depending on whether it is present in a given image or not) to obtain $\tilde{c}_t$ (representing intervention step $t \in \{1, ..., T\}$). This intervened concept representation is then passed into the concept realignment module (leveraging e.g. an MLP or LSTM reweighting mode), which outputs the realigned $u(\tilde{c}_t)$. To ensure that the ground-truth values provided by the user are not overwritten during realignment, $u(\tilde{c}_t)$ retains ground-truth corrections. The final concept vector is then based into a concept-based classifier $f$.

*intervention realignment module* (**CIRM**), comprising two interdependent components: (a) a *concept realignment model* (**CRM**), $u : \mathcal{C} \to \mathcal{C}$. After a user intervenes on a subset of concepts $\mathcal{S}$, the remaining concepts ($\backslash \mathcal{S}$) are realigned; and (b) an *intervention policy* $\pi$. The concepts predicted by the realignment model fed into the policy to suggest the next interventions. Both components are interdependent, and together form the overall concept intervention realignment module (Fig. 1).

The training of the full CIRM comprising both selection policy and concept realignment model aims to simulate the complete intervention process. As in §2.1, let $\mathcal{S}_t$ denote the set of intervened concepts and $\tilde{c}_t = \{c_{\mathcal{S}_t}, \hat{c}_{\backslash \mathcal{S}_t}\}$ denote the concepts at time $t$, respectively. At every intervention step, we feed $\tilde{c}_t$ to the realignment model to obtain updated concept predictions as $\kappa_t = u(\tilde{c}_t)$, which then are utilized by $\pi(\kappa_t)$ to produce intervention recommendations for $t + 1$. Finally, we train $u$ with ground-truth labels as targets using $\mathcal{L}(u) = (\sum_{t=0}^{T} \text{CE}(u(\tilde{c}_t), c))/T$. Further details in Supp. A.1.

## 4 EXPERIMENTS

**Implementation Details.** We conduct our experiments using Sequential CBMs and Concept-Embedding Models (CEMs; Zarlenga et al. (2022)) on the CUB (Wah et al., 2011), CelebA (Liu et al., 2015), and AwA2 (Xian et al., 2018) datasets. Our experimental setup closely follows Zarlenga et al. (2022) (note that for CelebA, we select the eight most balanced concepts, without any hidden concepts). Our overall training scheme consists of two phases. Similar to standard CBMs, we first train the backbone $f$ and the classification head $g$. Subsequently, we freeze those components and train the realignment model. Details regarding its training are available in Supp. A.2.

**Findings.** Across all datasets, we can observe a consistent, in parts vast reduction in concept prediction loss, which measures the correct assignment of concepts for each input (Fig. 2). For example on CUB, a *tenfold* reduction of the original uninterveined concept loss ($\sim 0.6$ to $\sim 0.06$) can be achieved with half the number of interventions (11 with concept realignment, 23 without). On top of that, we also find that the significant gain in concept attribution also translates to subsequent gains in intervention efficacy for the overall classification performance (Fig. 3). For example on CUB, the final classification accuracy after intervening on all concepts is $93.9\%$, which is achieved already after $\sim 16$ intervention steps. In Tab. 1, we show that these observations hold across a variety of CBMs and CEMs. Overall, our results highlight that concept realignment is crucial to best leverage human feedback in concept-based models at test-time, significantly reducing human effort.

Next, we study the effect of various design choices for the realignment module along two dimensions: **(1) Recurrent vs. Feedforward Networks:** Since we intervene on concepts sequentially, it is possible that the realignment module can benefit from the overall order and history of interventions to make more accurate concept predictions. To do this, we instantiate the concept realignment network using an LSTM Hochreiter & Schmidhuber (1997). We compare this against our default

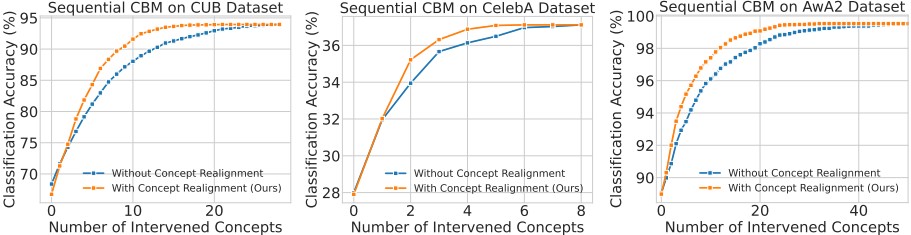

Figure 2: Concept prediction loss versus the number of intervened concepts with and without concept correction. Concept correction consistently improves concept predictions.

Figure 3: Classification accuracy versus number of intervened concepts with and without concept correction. Concept correction consistently improves classification accuracy.

MLP. **(2) Previous Output vs. Original Concepts:** By default, the realignment module takes as input a combination of ground-truth concepts provided by the user and values predicted by the base model at $t = 0$ for the concepts that have not been intervened on (see also §3). Due to the sequential nature of interventions, one may also directly feed the output of the realignment module at time $t-1$ as input to it at time $t$ in order to compound the refinements over multiple time steps. Combining both axes results in four recombinations, which we compare in Fig. 4. As can be seen, there is limited gain when accounting for the complete intervention history using an LSTM realigner network. Similarly, we find that applying the MLP primarily for concept selection alongside UCP and as final input to the classification head works better than compounding refinements over intervention steps.

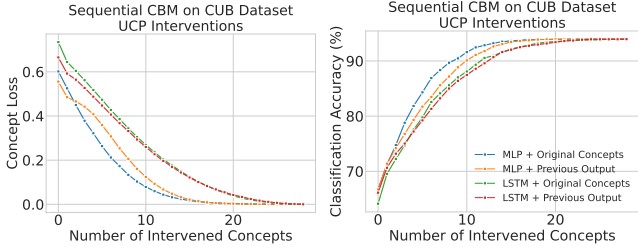

Figure 4: (a) Concept prediction loss and (b) classification accuracy for various realigner architectures alongside UCP policy. Using an MLP with concept predictions of the base model works better than compounding refinements and accounting for intervention trajectories using LSTMs.

## 5 DISCUSSION

In this work, we identify the independent treatment of concepts during test-time interventions in CBMs as a cause for low label efficiency and overall reduced intervention efficacy. To remedy this problem, we propose a simple and lightweight technique to automatically update all concepts after a human intervenes on a subset of concepts. Our experiments demonstrate that our approach enables for the selection of more relevant concepts, and consistently improves *both* concept and label predictions for the same number of human interventions. We believe that the consequently reduced cost of human intervention can facilitate the practical use of concept-based models, and encourage deployment even in resource-constrained environments.

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

# A  APPENDIX

## A.1  REALIGNMENT MODELS

As shown in Fig. 1, we parameterize our concept realignment model with a neural network $v$. To ensure that $u$ does not overwrite the ground-truth concepts provided by the user, we also keep track of the already intervened concepts $\mathcal{S}_t$. Using this information, we replace the output of the realigned concept embedding with the user-provided values for concepts in $\mathcal{S}_t$. Hence, the final output of $u$ for the $i^{th}$ concept is given as

$$u(\tilde{c}_t, \mathcal{S}_t)^{(i)} = \begin{cases} v(\tilde{c}_t)^{(i)} & \text{if } i \notin \mathcal{S}_t \\ \tilde{c}_t^{(i)} & \text{if } i \in \mathcal{S}_t. \end{cases}$$

Depending on the assumptions made on the realignment process, $v$ is either a simple MLP or a recurrent model (such as an LSTM Hochreiter & Schmidhuber (1997)). The former parametrizes our default concept intervention realignment model, which only passes the set of intervened and un-intervened concepts at intervention step $t$ to the concept realignment model consisting of a simple MLP. The set of concepts fed into the MLP may either be the original concept embedding $\tilde{c}_0$, where all intervened concepts up to and including step $t$ have been replaced with ground-truth values, or the previously realigned $\kappa_{t-1}$ with similarly updated intervened concepts (c.f. Fig. 1, "GT"). Note that in either case, $\kappa_{t-1}$ informs the selection process of the subsequent concept to intervene on. After all interventions, the final concept embedding fed into the classifier is always $u(\tilde{c}_T)$. Practically, we found using $\tilde{c}_t$ to work slightly better than $\kappa_{t-1}$. Both cases above however only pass the final set of concepts at time $t$ to the realignment model. Given the sequential nature of interventions, however, it may also be beneficial to account for the entire intervention history to inform future concept realignment. As a result, we also introduce a recurrent realignment variant, $u_{\text{rec}}$, which employs an LSTM model to retain the entire history of interventions until time $t$.

## A.2  IMPLEMENTATION DETAILS

We perform experiments on CEMs, IntCEMs, and three types of CBMs (sequential, independent, and joint). For all models and datasets, we follow the hyperparameters used in Zarlenga et al. (2023). During CIRM training, we sequentially intervene on concepts $T = k$ times. By default, we use UCP both during training and inference, and if not stated otherwise, use a multi-layered perceptron (MLP) for concept realignment. We use the predictions of the base CBM ($\tilde{c}_t$) as its input for un-intervened concept representations. We perform a small, standard hyperparameter using Optuna Akiba et al. (2019) with 50 trials to search over the number of hidden layers $\in \{1, 2, 3\}$ and units $\in \{k, 2k, k/2\}$, the learning rate $\in [10^{-5}, 10^{-1}]$ and weight decay $\in [10^{-6}, 5 \times 10^{-5}]$, and use the same batch size as used to train the base model. We employ early stopping and learning rate decay on the validation loss. For joint training, we instantiate the realigner MLP 2 hidden layers containing $k$ neurons each. Experiments are conducted using PyTorch.

## A.3  ADDITIONAL RESULTS

Table 1: Area Under Curve (AUC) of Concept Prediction Loss and Classification Accuracy with/without CIRM. We use the same backbone for sequential and independent CBMs. CIRM improves performance across all models and datasets. Intervention curves share long saturation plateaus for high intervention counts. Accuracy AUC scores are thus saturated, and best combined with performance graphs in Figs. 2, 3.

| Base Model | Realigned | Concept Loss AUC ↓ | | | Accuracy AUC ↑ | | |
|---|---|---|---|---|---|---|---|
| | | CUB | CelebA | AwA2 | CUB | CelebA | AwA2 |
| Sequential CBM | × | 6.71 | 1.59 | 4.26 | 2460.8 | 280.7 | 8364.0 |
| | ✓ | **3.15** | **1.52** | **1.13** | **2510.9** | **284.3** | **8397.6** |
| Independent CBM | × | 6.71 | 1.59 | 4.26 | 2653.4 | 280.2 | 8403.4 |
| | ✓ | **3.15** | **1.52** | **1.13** | **2678.3** | **282.1** | **8437.0** |
| Joint CBM | × | 5.93 | 3.06 | 4.77 | 2580.3 | 273.1 | 8276.4 |
| | ✓ | **3.67** | **1.76** | **1.48** | **2609.0** | **273.9** | **8327.4** |
| CEM | × | 5.99 | 1.61 | 4.90 | 2521.4 | 396.3 | 8429.3 |
| | ✓ | **3.20** | **1.46** | **1.69** | **2558.4** | **400.1** | **8433.9** |

