# OpenReview forum: "Improving Intervention Efficacy via Concept Realignment in Concept Bottleneck Models"
_ICLR.cc/2024/Workshop/Re-Align — ICLR 2024 Workshop Re-Align Poster_

### Official Review · Reviewer_4Wv1 · 2024-02-23
**Getting More from a Single Intervention in Concept-Based Systems**

**Rating:** 2
**Fit:** 2
**Confidence:** 2

**Workshop Review:**

Concept-based systems are intervenable by design, enabling humans to directly modify models' intermediate predictions over concepts. Yet, querying humans to intervene on many concepts can be costly. Prior work has studied and developed policies which strive to select a minimal set of concepts which are most valuable for humans to intervene on. For each concept intervention received from a human, these works typically only edit one concept a time. However, the concept spaces over which such models are trained often have richer interdependencies, which if leveraged, could lead to more efficient interventions -- sharing information from a single human intervention across multiple concept values. It is this problem which the authors focus on.

The authors design a new "concept correction" method which leverages the correlational structure over concepts to improve update efficiency. The authors validate their method across two classes of concept-based models (CBMs and CEMs). The results are promising and certainly a good work-in-progress.

As the authors continue to build out the work, I'd suggest exploring a 2x2 cross-product (or more). Right now, the authors seem to be varying two things: the policy (static vs. dynamic) and the correction measure (non-shared vs. their proposed stat-based sharing). The authors only investigate 3 of these 4 combinations. It would be good to run all four, e.g., concept correction interventions + random?

It was also not clear to me how the authors are thinking about framing their work more broadly; is the method being proposed a new intervention policy or way of handling the interventions that come back for any given policy? Both are important, but proposing something new on the former needs more eval and comparison to baselines. I’d suggest prioritizing eval on the how-interventions-are-handled question (rather than exploring new intervention policies, per say), but look forward to seeing what the authors come up with.

A few minor notes:
- The figures were generally hard to read. What are the error bars?
- Typo in abstract – adwi … addition
- Some potentially important cites where missing. Particularly around the challenges of humans for providing good concept interventions, e.g., "Overlooked factors in concept-based explanations: Dataset choice, concept salience, and human capability" and "Human uncertainty in concept-based AI systems." Along with some other work that looks at reducing concept sizes "Selective Concept Models: Permitting Stakeholder Customisation at Test-Time".

**Reason For Not Giving Higher Score:**

I thought the work was a very good work-in-progress; however, I found the figures were not quite clear and the authors didn't cover the full space of variants of their correction and policy (per note above). It also wasn't entirely clear to me how the authors are pitching their method wrt a new intervention policy and/or new method that can interlink with an existing policy.

For this workshop in particular, I felt the authors missed an opportunity to make a clearer connection to human-model representation alignment (in concept space) that would have increased topical fit.

**Reason For Not Giving Lower Score:**

The work is timely, and the authors' idea around having a concept correction factor is well-motivated and has potential broader impact to improve the usability of concept-based systems writ large.

**Reviewer Domain:**

machine learning

---

### Official Review · Reviewer_MMei · 2024-02-23
**Interesting idea: intervention correction for CBM**

**Rating:** 2
**Fit:** 2
**Confidence:** 2

**Workshop Review:**

Overall, the workshop paper presents a novel idea clearly, with thorough testing and motivation. Consequently, I would recommend accepting it. However, there are some major concerns (**M**) and minors (**m**) that I'd like to point out:

**M.1**: One significant question arises regarding the choice of accuracy as a metric. CBMs were initially designed for interpretability rather than accuracy. Thus, the integration of interventions to enhance accuracy raises concerns about potentially downside in interpretability. Additionally, the interpretability of interventions becomes questionable as they are subsequently adjusted by $u(\cdot)$. If human interventions diverge significantly from statistical relationships, $u(\cdot)$ may counteract these interventions, resulting in interventions that are entirely uninterpretable.

As for minor issues:

**m.1**: The plot depicting accuracy versus loss seems redundant.

**m.2**: It would be beneficial to include images before and after correction to visually demonstrate the effect of $u(\cdot)$. A clearer explanation of how $u(\cdot)$ re-aligns or fixes interventions would enhance understanding.

**m.3**: For future versions of the paper, it would be valuable to conduct a study where humans assess the modifications made by $u(\cdot)$ to gauge their coherence. Additionally, providing insights into the statistical relationships captured by $u(\cdot)$ would be insightful for understanding the underlying processes (co-occuring concept...).

**m.4**:  I recommend including the following references:
- Sawada & Nakamura (2022) - CBM-AUC
- Alvarez-Melis & Jaakkola (2018) - Self-Explaining Neural Networks (CBM before CBM)
- Stammer et al. (2022) - Method for Learning Concepts with Weaker Supervision
- Mahinpei et al. (2021), Margeloiu et al. (2021), Havasi et al. (2022) - Concept Leakage
- Lockhart et al. (2022) - Concept Bottleneck Models with Dropped Concept Predictions

**Reason For Not Giving Higher Score:**

See **M.1**

**Reason For Not Giving Lower Score:**

The paper is clearly motivated, pointed out a real problem in intervention of CBM and is well-written.

**Reviewer Domain:**

machine learning

---

### Official Review · Reviewer_cpRq · 2024-03-01
**Idea is good but presentation need to be improved and more details regarding experiments would be helpful**

**Rating:** 1
**Fit:** 3
**Confidence:** 1

**Workshop Review:**

The paper works on the problem of correlated concept intervention for CBMs (Concept Bottleneck Models), in the hope to reduce the number of interventions required.

The authors proposed and developed a concept corrector model u which is used to re-align non-intervened concepts based on learned statistical concept relationships. The u is trained on the same training data used by the CBM, by starting from the concept predictions of the base model, and sequentially intervening on concepts for k time-steps. The u is trained on binary cross-entropy loss.

While the idea sounds appealing and Figure 1,2,3,4 show positive preliminary results. The paper presentation needs substantial improvements to add more details and clear to follow.

**Reason For Not Giving Higher Score:**

The communication/writing of the method used and how it instantiates in both of the dataset is unclear. For example, for celebA dataset, it is useful to show what concepts are modified, what kind of statistical relationship is learned by concept corrector.

Misc:
Typo in abstract: 1st line CMB->CBM; adwdition->addition

**Reason For Not Giving Lower Score:**

The idea sounds feasible and appealing.

**Reviewer Domain:**

machine learning

---

### Decision · Program_Chairs · 2024-03-02

Accept (Poster)